# The Dual Nature of Chaos and Order in the Atmosphere

**Bo-Wen Shen [1,*] , Roger Pielke, Sr. [2], Xubin Zeng [3] , Jialin Cui [4], Sara Faghih-Naini [5,6], Wei Paxson [1], Amit Kesarkar [7], Xiping Zeng [8] and Robert Atlas [9,†]**

1   Department of Mathematics and Statistics, San Diego State University, San Diego, CA 92182, USA
2   Cooperative Institute for Research in Environmental Sciences, University of Colorado Boulder, Boulder, CO 80203, USA
3   Department of Hydrology and Atmospheric Science, The University of Arizona, Tucson, AZ 85721, USA
4   Department of Computer Science, North Carolina State University, Raleigh, NC 27695, USA
5   Department of Mathematics, University of Bayreuth, 95447 Bayreuth, Germany
6   Department of Computer Science, Friedrich-Alexander University Erlangen-Nuremberg, 91058 Erlangen, Germany
7   National Atmospheric Research Laboratory, Gadanki 517112, India
8   Army Research Laboratory, Adelphi, MD 20783, USA
9   Atlantic Oceanographic and Meteorological Laboratory, National Oceanic and Atmospheric Administration, Miami, FL 33149, USA
*   Correspondence: bshen@sdsu.edu
†   The author has retired.

**Abstract:** In the past, the Lorenz 1963 and 1969 models have been applied for revealing the chaotic nature of weather and climate and for estimating the atmospheric predictability limit. Recently, an in-depth analysis of classical Lorenz 1963 models and newly developed, generalized Lorenz models suggested a revised view that "*the entirety of weather possesses a dual nature of chaos and order with distinct predictability*", in contrast to the conventional view of "*weather is chaotic*". The distinct predictability associated with attractor coexistence suggests limited predictability for chaotic solutions and unlimited predictability (or up to their lifetime) for non-chaotic solutions. Such a view is also supported by a recent analysis of the Lorenz 1969 model that is capable of producing both unstable and stable solutions. While the alternative appearance of two kinds of attractor coexistence was previously illustrated, in this study, multistability (for attractor coexistence) and monostability (for single type solutions) are further discussed using kayaking and skiing as an analogy. Using a slowly varying, periodic heating parameter, we additionally emphasize the predictable nature of recurrence for slowly varying solutions and a less predictable (or unpredictable) nature for the onset for emerging solutions (defined as the exact timing for the transition from a chaotic solution to a non-chaotic limit cycle type solution). As a result, we refined the revised view outlined above to: "*The atmosphere possesses chaos and order; it includes, as examples, emerging organized systems (such as tornadoes) and time varying forcing from recurrent seasons*". In addition to diurnal and annual cycles, examples of non-chaotic weather systems, as previously documented, are provided to support the revised view.

**Keywords:** dual nature; chaos; generalized Lorenz model; predictability; multistability

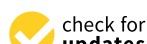



## 1. Introduction

Two studies of Prof. Lorenz (Lorenz 1963, 1972 [1–3]) laid the foundation for chaos theory that emphasized the feature of Sensitive Dependence of Solutions on Initial Conditions (SDIC) [4,5]. Such a feature can be found in earlier studies [6–8] and was rediscovered by Lorenz [1]. Since Lorenz's studies in the 1960s, the concept of SDIC has changed the view on the predictability of weather and climate, yielding a paradigm shift from Laplace's view of determinism with unlimited predictability to Lorenz's view of deterministic chaos with finite predictability. Specifically, the statement "weather is chaotic" has been well accepted [9,10].

Amongst Lorenz models [1,2,11–16], the Lorenz 1963 and 1969 models have been applied to reveal the chaotic nature of the atmosphere and to estimate a predictability horizon. While the Lorenz 1963 (L63) equations represent a limited-scale, nonlinear, chaotic model [1,3,4], the Lorenz 1969 (L69) model is a closure-based, physically multiscale, mathematically linear, and numerically ill-conditioned system [17]. Although the L63 and L69 models have been applied for revealing the chaotic nature and for estimating the predictability limit for the atmosphere, we, in the scientific community, should be fully aware of the strength and weakness of the idealized models. As reviewed in Section 2, both models also produce various types of solutions. However, in past years, only chaotic solutions of the L63 model and linearly unstable solutions of the L69 model have been applied for studying atmospheric predictability. Therefore, asking whether or not other features of the classical Lorenz models and new features of generalized Lorenz models can improve our understanding of the nature of weather and climate is legitimate.

In contrast to single type chaotic solutions applied to reveal the nature of the atmosphere, recent studies using classical, simplified, and generalized Lorenz models [18–20] have emphasized the importance of considering various types of solutions, including solitary type solutions and coexisting chaotic and non-chaotic solutions, for revealing the complexity of the atmosphere and, thus, distinct predictability. For example, the physical relevance of findings within Lorenz models for real world problems has been reiterated by providing mathematical universality between the Lorenz simple weather and the Pedlosky simple ocean models [21–24]; and amongst the non-dissipative Lorenz model, the Duffing, the Nonlinear Schrodinger, the Korteweg–de Vries equations, and the simplified, epidemic SIR model [25–29]. Such an effort illustrated that mathematical solutions of the classical Lorenz model can help us understand the dynamics of different physical phenomena (e.g., solitary waves, homoclinic orbits, epidemic waves, and nonlinear baroclinic waves).

The L63 model mainly produces single types of nonlinear solutions, although it yields coexisting attractors over a very small interval for the heating parameter (e.g., [30]). By comparison, a newly developed generalized Lorenz model (GLM) with nine state variables yields attractor coexistence over a wide range of parameters. Such a feature is defined when chaotic and non-chaotic solutions appear within the same GLM that applies the same model parameters but different initial conditions. Additionally, when a time varying heating function is applied, various types of solutions may alternatively and concurrently appear. Based on findings obtained from classical and generalized Lorenz models [9,10], the conventional view is being revised to emphasize the dual nature of chaos and order, as follows:

"Weather possesses chaos and order; it includes, as examples, emerging organized systems (such as tornadoes) and time varying forcing from recurrent seasons".

Since 2019 [9,10,18–20], the first portion of the above quote has been explicitly documented in our recent studies using the L63 model and GLM, while the second portion of the quote was implicitly suggested in Shen et al., 2021 [9]. In addition to various types of solutions within Lorenz models, earlier studies that applied regular solution to study non-chaotic weather systems were summarized in Shen et al., 2021 [9,10]. To support the revised view with a focus on the illustration for the second sentence, this article not only provides a brief review but also elaborates on additional details for the following features: (1) an analogy for monostability and multistability using skiing vs. kayaking; (2) single-types of attractors, SDIC, and monostability within the L63 model; (3) coexisting attractors and multistability within the GLM; (4) time varying multistability and slow time varying solutions; (5) the onset of emerging solutions; (6) various types of solutions within the L69 model; (7) distinct predictability within Lorenz models; (8) a list of non-chaotic weather systems; and (9) a short list of suggested future tasks. Table 1 lists the definitions of related concepts.

**Table 1.** The definitions of concepts related to predictability and multistability.

| Name | Definitions | Recommendations |
|---|---|---|
| 1st kind of attractor coexistence | The coexistence of chaotic and steady-state solutions | [9,10,28] |
| 2nd kind of attractor coexistence | The coexistence of nonlinear oscillatory and steady-state solutions | [9,10] |
| attractor | The smallest attracting point set that, itself, cannot be decomposed into two or more subsets with distinct basins of attraction. | [31] |
| autonomous | A system of ODEs is autonomous if time does not explicitly appear within the equations. | [32] |
| bifurcation | It occurs when the structure of a system's solution significantly changes as a control parameter varies. | [32,33] |
| butterfly effect | *The phenomenon that a small alteration in the state of a dynamical system will cause subsequent states to differ greatly from the states that would have followed without the alteration.* | [3] |
| basin of attraction | As time advances, orbits initialized within a basin tend asymptotically to the attractor lying within the basin. The set of initial conditions leading to a given attractor. | [33] |
| chaos | Bounded aperiodic orbits exhibit a sensitive dependence on ICs. | [3] |
| final state sensitivity | Nearby orbits settle to one of multiple attractors for a finite but arbitrarily long time. | [34] |
| hidden attractor | An attractor is called a hidden attractor if its basin of attraction does not intersect with small neighborhoods of equilibria. | [35] |
| intransitivity | A specific type of solution lasts forever. | [36] |
| intrinsic predictability | Predictability that is only dependent on flow itself. | [9,37] |
| limit cycle | A nonlinear oscillatory solution; an isolated closed orbit | [32] |
| monostability | The appearance of single-type solutions | [9,10,17] |
| multistability | A system with multistability contains more than one bounded attractor that depends only on initial conditions. For example, the coexistence of two types of solutions. | [9,10,17,38,39] |
| non-autonomous | Variable time ($\tau$) appears on the right-hand side of the equations. | [32] |
| phase space | Within a system of the first-order ODEs, a phase space or state space can be constructed using time-dependent variables as coordinates. | [40] |
| practical predictability | Predictability that is limited by imperfect initial conditions and/or (mathematical) formulas. | [9,37] |
| recurrence | Defined when a trajectory returns back to the neighborhood of a previously visited state. Recurrence may be viewed as a generalization of "periodicity" that braces quasi-periodicity with multiple frequencies and chaos. | [33] |
| sensitive dependence | *The property characterizing an orbit if most other orbits that pass close to it at some point do not remain close to it as time advances.* | [3] |

## 2. Analysis and Discussion

As outlined at the end of Section 1, this section first applies skiing and kayaking to provide an analogy for monostability and multistability. The section then discusses major features of the L63 model, the GLM, and the L69 model. To present monostability and multistability, various types of solutions are reviewed. A list for non-chaotic weather systems is provided at the end of the section.

### 2.1. An Analogy for Monostability and Multistability Using Skiing and Kayaking

Since the SDIC, monostability, and multistability are the most important concepts in this study, to help readers, they are first illustrated using real-world analogies of skiing and kayaking. To explain SDIC, the book entitled "The Essence of Chaos" by Lorenz, 1993 [3] applied the activity of skiing (left in Figure 1) and developed an idealized skiing model for revealing the sensitivity of time-varying paths to initial positions (middle in Figure 1). Based on the left panel, when slopes are steep everywhere, SDIC always appears. This feature with a single type of solution is referred to as monostability.

**Figure 1.** Skiing as used to reveal monostability (left and middle, Lorenz 1993 [3]) and kayaking as used to indicate multistability (right, courtesy of Shutterstock-Carol Mellema https://www.shutterstock.com/image-photo/kayaker-enjoys-whitewater-sinks-smoky-mountains-649533271 (accessed 1 November 2022)). A stagnant area is outlined with a white box.

In comparison, the right panel of Figure 1 for kayaking is used to illustrate multistability. In the photo, the appearance of strong currents and a stagnant area (outlined with a white box) suggests instability and local stability, respectively. As a result, when two kayaks move along strong currents, their paths display SDIC. On the other hand, when two kayaks move into a stagnant area, they become trapped, showing no typical SDIC (although a chaotic transient may occur [31]). Such features of SDIC or no SDIC suggest two types of solutions and illustrate the nature of multistability.

## 2.2. Single-Types of Attractors, SDIC, and Monostability within the L63 Model

The well-known L63 model that consists of three, first order ordinary differential equations (ODEs) with three state variables and three parameters is written as follows [1,41]:

$$\frac{dX}{d\tau} = \sigma Y - \sigma X, \tag{1}$$

$$\frac{dX}{d\tau} = \sigma Y - \sigma X, \tag{2}$$

$$\frac{dZ}{d\tau} = XY - bZ. \tag{3}$$

Here, $\tau$ is dimensionless time. Time-independent parameters include $\sigma$, $r$, and $b$. The first two parameters are known as the Prandtl number and the normalized Rayleigh number (or the heating parameter), respectively. The heating parameter represents a measure of temperature differences between the bottom and top layers, while the third parameter, $b$, roughly indicates the aspect ratio of the convection cell. This study keeps $\sigma = 10$ and $b = 8/3$, but varies $r$ in different runs. For the three state variables, $X$, $Y$, and $Z$, the variable $X$ is associated with a stream function that defines velocities, and variables $Y$ and $Z$ are related to temperature. The 2-page Supplementary Materials of Shen et al., 2021 [9] list the equations of the GLM that can be reduced to the L63 model. At small, medium, and large heating parameters, three types of solutions are known to appear [9,10,19]. Below, three numerical experiments, which apply three different values of Rayleigh parameters and keep the other two parameters as constants within the same L63 model, are discussed. The selected values of the Rayleigh parameters are 20, 28, and 350, representing weak, medium, and strong heating, respectively. For each of the three cases, both control and parallel runs were performed. The only difference in the two runs was that a tiny perturbation with $\epsilon = 10^{-10}$ was added into the initial condition (IC) of the parallel run. The SDIC along with continuous dependence of solutions on IC (CDIC) is then discussed. Solutions of the $Y$ component are provided in the top panels of Figure 2. Three panels, from left to right, display long-term time independent responses, irregular temporal variations, and regular temporal oscillations referred to as steady-state, chaotic, and limit cycle solutions, respectively. Corresponding solutions within the $X$-$Y$ phase space are shown in the bottom panels of Figure 2. Non-chaotic, steady-state and limit cycle solutions become point attractors and periodic attractors in panels 1d and 1f, respectively. As a comparison, all of the three types of attractors within the three-dimensional $X$-$Y$-$Z$ phase space are provided in Figure 3. Below, the features of SDIC for chaotic solutions are further discussed.

For chaotic solutions in the middle panels of Figure 2, both the control and parallel runs produced very close responses at an initial stage, but very different results at a later time. Initial comparable results indicate that CDIC is an important feature of dynamical systems. Despite initial tiny differences, large differences at a later time, as indicated by the red and blue curves in Figure 2b, revealed the feature of SDIC. Such a feature suggests that a tiny change in an IC will eventually lead to a very different time evolution for a solution. However, the concept of SDIC does not suggest a causality relationship. Specifically, the initial tiny perturbation should not be viewed as the cause for a specific event (e.g., a maximum or minimum) that subsequently appears or for any transition between different events.

As discussed above, depending on the relative strength of the heating parameter, one and only one type of steady-state, chaotic, and limit cycle solution appears within the L63 model. Such a feature is referred to as monostability, as compared to multistability for coexisting attractors. Over several decades, chaotic solutions and monostability have been a focal point, yielding the statement "weather is chaotic". As discussed below, such an exclusive statement is being revised by taking coexisting attractors and time varying multistability into consideration.

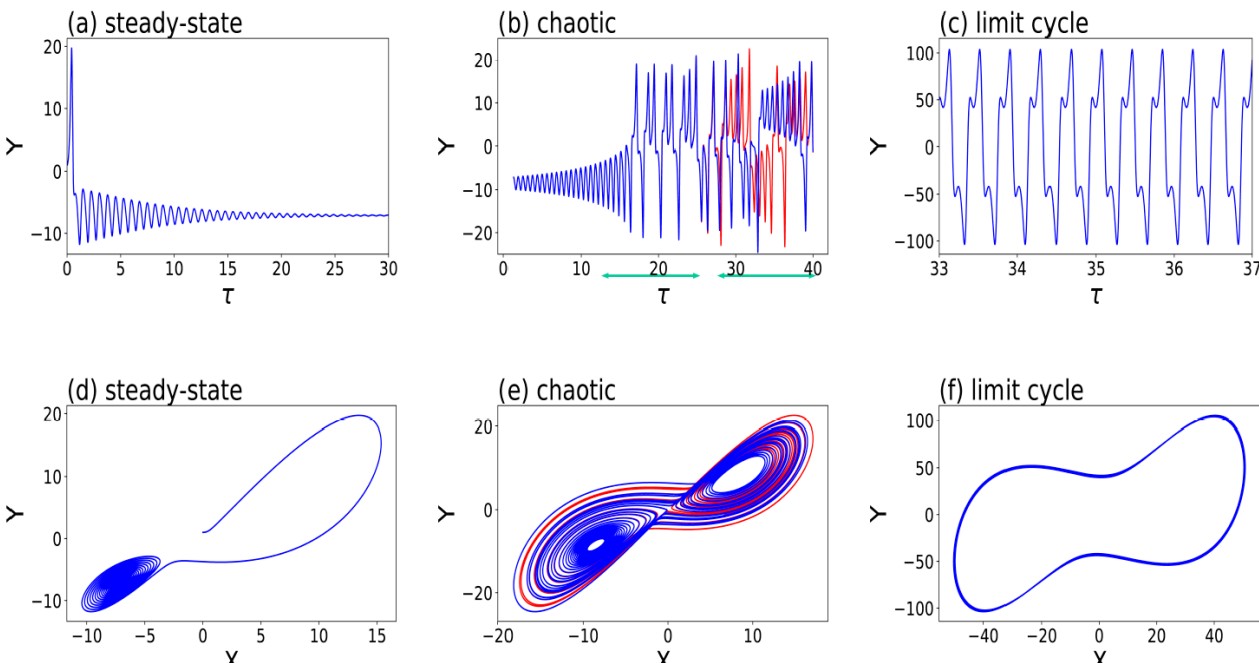

**Figure 2.** Three types of solutions within the Lorenz 1963 model. Steady-state (**a**,**d**), chaotic (**b**,**e**), and limit cycle (**c**,**f**) solutions appear at small, moderate, and large normalized Rayleigh parameters (i.e., *r* = 20, 28, and 350), respectively. Control and parallel runs are shown in red and blue, respectively. SDIC is indicated by visible blue and red curves in panel (**b**), where the first and second green horizonal lines indicate CDIC and SDIC, respectively. (**a**–**c**) depict the time evolution of Y. (**d**–**f**) show orbits within the *X–Y* space, appearing as a point attractor (**a**,**d**), a chaotic attractor (**b**,**e**), and a periodic attractor (**c**,**f**), respectively (after Shen et al., 2021 [10]). The other two parameters are kept as constants: σ = 10 and b = 8/3. The initial conditions of $(X, Y, Z)$ for the control and parallel runs are $(0, 1, 0)$ and $(0, 1 + \epsilon, 0)$, respectively.

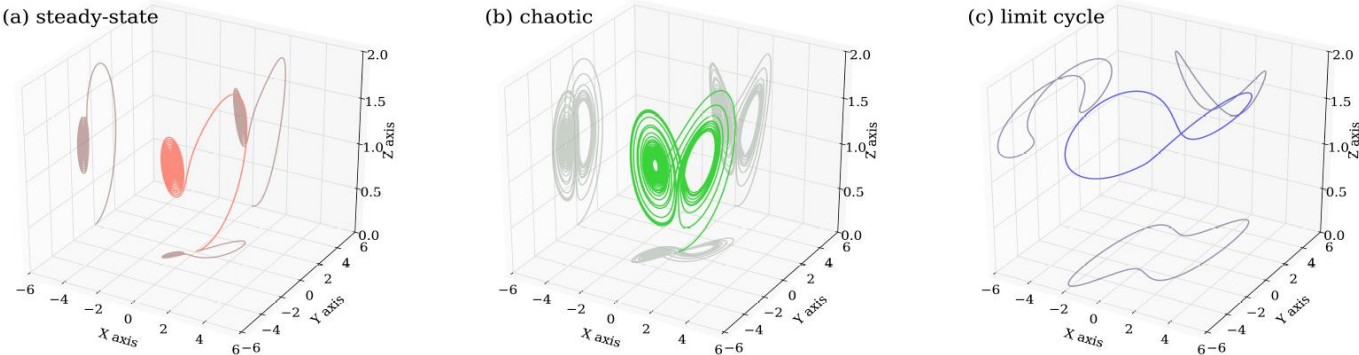

**Figure 3.** Three types of solutions within the *X–Y–Z* phase space obtained from the Lorenz 1963 model. Panels (**a**–**c**) display a steady-state solution, a chaotic solution, and a limit cycle with small, medium, and large heating parameters, respectively. While panels (**a**,**b**) show the solution for τ ∈ [0, 30] panel, to reveal its isolated feature, panel (**c**) displays the limit cycle solution for τ ∈ [10, 30]. Values of parameters are the same as those in the control run in Figure 2.

### 2.3. Coexisting Attractors and Multistability within the GLM

Based on various high-dimensional Lorenz models [41–44], a generalized multi-dimensional Lorenz model (GLM) has been developed [18–20]. Mathematical descriptions of the GLM are summarized in the Supplementary Materials of Shen et al., 2021 [9]. Major features of the GLM include: (1) any odd number of state variables greater than three; (2) the aforementioned three types of solutions; (3) hierarchical spatial scale dependence (e.g., [43,44]); and (4) two kinds of attractor coexistence [9,18,20,45,46]. Additionally, aggregated negative feedback appears within high-dimensional LMs when the negative feedback of various smaller scale modes is accumulated to provide stronger dissipations, requiring stronger heating for the onset of chaos in higher-dimensional LMs. Such a finding is indicated in Table 2 of Shen, 2019 [19], which compared the critical values of heating parameters for the onset of chaos within the L63 and GLM that contains 5–9 state variables. Sufficiently large, aggregated negative feedback may cause (some) unstable equilibrium points to become stable and, thus, enable the coexistence of stable and unstable equilibrium points, yielding attractor coexistence, as illustrated below using the GLM with nine state variables.

Amongst the three types of solutions (i.e., steady-state, chaotic, and limit-cycle solutions), two types of solutions may appear within the same model that applies the same model parameters but different initial conditions. Such a feature is known as attractor coexistence. The GLM produces two kinds of attractor coexistence, including coexisting chaotic and steady-state solutions and coexisting limit-cycle and steady-state solutions, referred to as the 1st and 2nd kinds of attractor coexistence, respectively. To illustrate coexisting attractors that display a dependence on initial conditions, Figure 4 displays two ensemble runs, each run using 128 different initial conditions that were distributed over a hypersphere (e.g., Shen et al., 2019 [20]). The 1st and 2nd kinds of attractor coexistence are illustrated using values of 680 and 1600 for the heating parameter, respectively. As shown in Figure 4a, 128 orbits with different starting locations eventually reveal the 1st kind of attractor coexistence for one chaotic attractor, and two point attractors. As clearly seen in Figure 4a, each of the chaotic and non-chaotic attractors occupy a different portion of the phase space. By comparison, in Figure 4b, 128 ensemble members produce the 2nd kind of attractor coexistence, consisting of limit cycle and steady-state solutions.

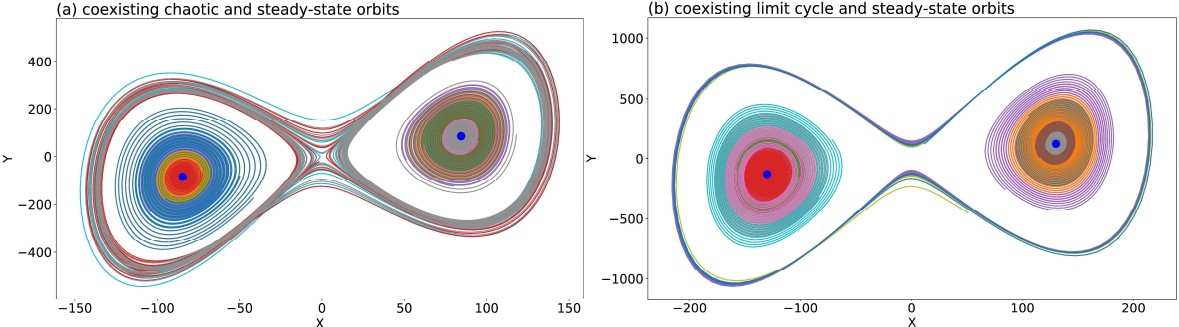

**Figure 4.** Two kinds of attractor coexistence using the GLM with 9 modes. Each panel displays orbits from 128 runs with different ICs for $\tau \in [0.625, 5]$. Curves in different colors indicate orbits with different initial conditions. (**a**) displays the coexistence of chaotic and steady-state solutions with $r = 680$. Stable critical points are shown with large blue dots. (**b**) displays the coexistence of the limit cycle and steady-state solutions with $r = 1600$.

The above two kinds of attractor coexistence occur in association with the coexistence of unstable (i.e., a saddle point) and stable equilibrium points. As discussed in Shen, 2019 [18] and reviewed above, the appearance of local stable equilibrium points is enabled by the so-called aggregated negative feedback of small-scale convective processes. The feature with coexisting attractors is referred to as multistability, as compared to monostability for single type attractors. As a result of multistability, SDIC does not always appear. Namely, SDIC appears when two orbits become the chaotic attractor that occupies one

portion of the phase space; it does not appear when two orbits move towards the same point attractor that occupies the other portion of the phase space.

In a recent study by Shen et al. [47], three major kinds of butterfly effects can be identified: (1) SDIC, (2) the ability of a tiny perturbation in creating an organized circulation at a large distance, and (3) the hypothetical role of small scale processes in contributing to finite predictability. While the first kind of butterfly effect with SDIC is well accepted, the concept of multistability suggests that the first kind of butterfly effect does not always appear.

### 2.4. Time Varying Multistability and Recurrent Slowly Varying Solutions

Within Lorenz models, a time varying heating function may be applied to represent a large-scale forcing system ([9,36]; and references therein). Since the heating function changes with time, the first and second kinds of attractor coexistence alternatively appear, leading to time varying multistability and transitions from chaotic to non-chaotic solutions. An analysis of the above two features can help reveal the predictable nature of recurrence for slowly varying solutions, and a less predictable (or unpredictable) nature for the onset for emerging solutions (defined as the exact timing for the transition from a chaotic solution to a non-chaotic limit cycle type solution).

Here, by extending the study of Shen et al., 2021 [9], the GLM with a time-dependent heating function is applied in order to revisit time varying multistability. Figure 5a displays three trajectories during a dimensionless time $\tau$ between 0 and $35\pi$ (i.e., $\tau \in [0, 35\pi]$). As shown in the second panels of Figure 5, these solutions were obtained using tiny differences in ICs and a time varying Rayleigh parameter. The 3rd to 5th panels display the feature of SDIC [3,17,47], the first kind of attractor coexistence (i.e., coexisting steady-state and chaotic solutions), and the second kind of attractor coexistence (i.e., coexisting steady-state and periodic solutions) at different time intervals, respectively. The alternative appearance of two kinds of attractor coexistence suggests time varying multistability. By comparison, the 6th panel indicates that once a "steady state" solution appears (when local time changes for all state variables become zero), it remains "steady" and slowly varies with the time-dependent heating function. Such a recurrent, slowly varying solution may, in reality, be used as an analogy for recurrent seasons.

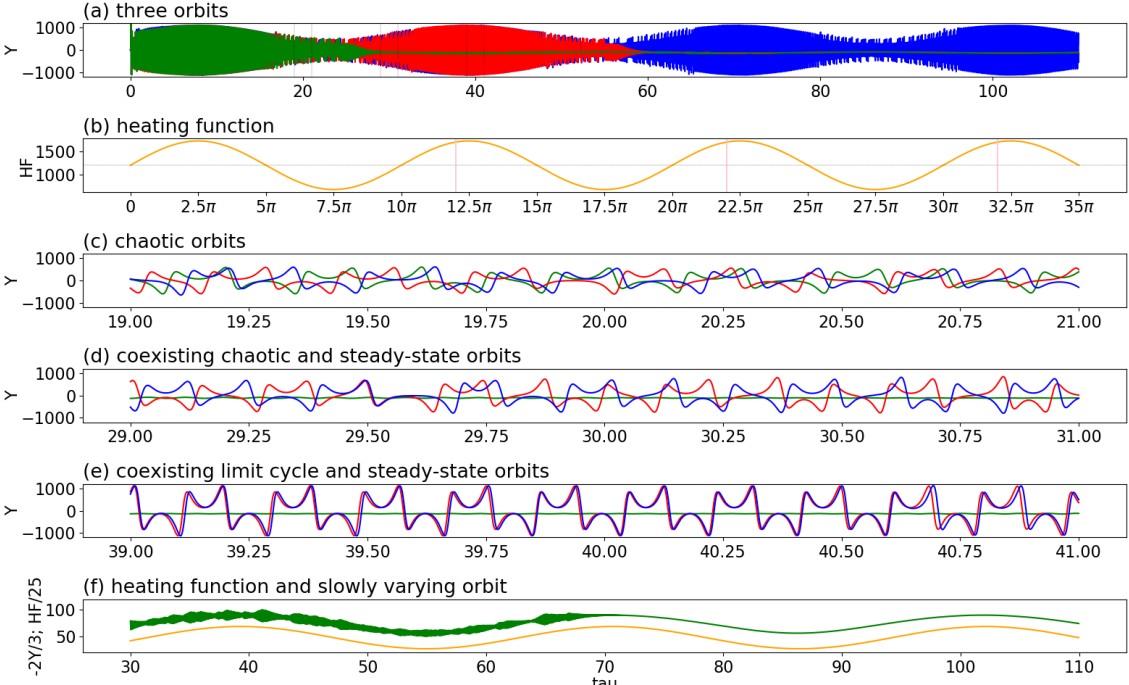

**Figure 5.** Two kinds of attractor coexistence revealed by three trajectories using a time varying heating parameter (i.e., Rayleigh parameter), $r = 1200 + 520 \sin (\tau/5)$, within a GLM (Shen, 2019 [18]).

The green, blue, and red lines represent the solutions of the control and two parallel runs. The parallel runs include an initial tiny perturbation, $\epsilon = 10^{-8}$ or $\epsilon = -10^{-8}$. The heating function is indicated by an orange line. From top to bottom, panels (**a**,**b**) display the three orbits and the heating parameters for $\tau \in [0, 35\pi]$, respectively. Panel (**c**) for $\tau \in [19, 21]$ displays diverged trajectories, showing SDIC. The first kind of attractor coexistence (i.e., coexisting chaotic and steady-state solutions) is shown in panel (**d**) for $\tau \in [29, 31]$. The green line, indeed, represents a steady state solution. The second kind of attractor coexistence (i.e., coexisting regular oscillations and steady-state solutions) is shown in panel (**e**) for $\tau \in [39, 41]$. Panel (**f**) displays a nearly steady-state solution (2Y/3) and the heating function for $\tau \in [30, 110]$. The three vertical lines in panel (**b**) indicate the starting time for the analysis in Figure 6. (After Shen et al., 2021 [9]).

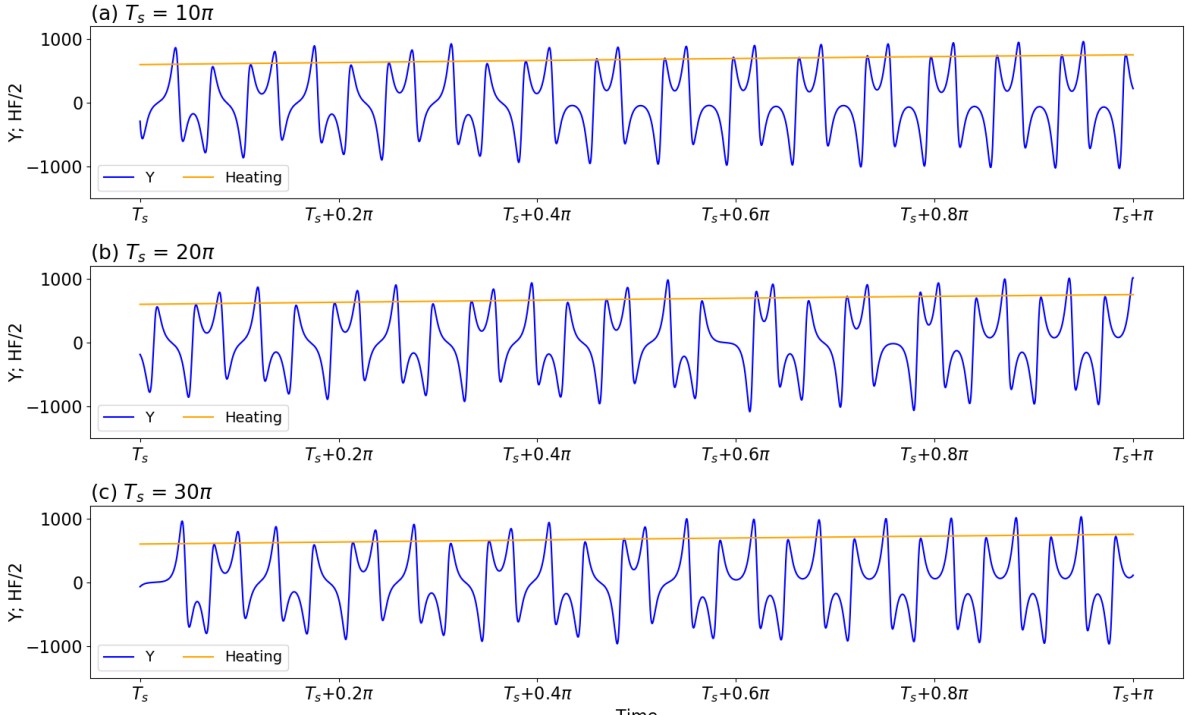

**Figure 6.** Panels (**a**–**c**) display the same trajectory during three different time intervals of $\tau \in [Ts, Ts + \pi]$, with the starting time *Ts* equal to $10\pi$, $20\pi$, and $30\pi$, respectively. The orange line in each panel represents the half value of the heating function.

### 2.5. Onset of Emerging Solutions

Although the GLM with a time dependent heating function produces time varying multistability, as a result of simplicity within the model and the complexity of the problem, here, no attempt is made to address the attractor basin, intra-transitivity, final state sensitivity, or a hidden attractor (e.g., [9,46]; see details in Table 1). Instead, the transition from one type of solution to the other type of solution is a focus. Here, for emulating a real-world scenario for the first appearance of African Easterly waves during a seasonal transition [9,19], the transition from a chaotic (irregular) solution to a limit cycle type (regular) solution is analyzed. Within Figure 5, while two trajectories display a sensitivity to initial conditions after $\tau > 19$ (Figure 5c), they become regularly oscillatory solutions with comparable frequencies and amplitudes for $\tau \in [39, 41]$ (Figure 5e). Reappearance of the regular solution (i.e., limit cycle type solutions) is defined as the "onset of an emerging solution". Below, a challenge in predicting the onset of the transition from a chaotic solution to a regularly oscillatory solution is illustrated.

Figure 6 displays the same trajectory during three different time intervals of $\tau \in [T_s, T_s + \pi]$. Here, the starting time $T_s$ is equal to $10\pi$, $20\pi$, and $30\pi$ in panels (a)–(c), respectively. An orange line in each panel represents the half value of the heating function. Given any vertical line, its intersection with each of the three orange lines in

all panels yields the same value for the heating parameter. In other words, although the starting time is different, the time evolution for the heating functions is exactly the same in all three panels. As a result, if the appearance of regularly oscillatory solutions is solely determined by the values of the heating function, the same time evolution of oscillatory solutions should appear in the three panels of Figure 6. However, differences are observed. For example, panel (a) displays the onset of a regularly oscillatory solution (e.g., the timing of the regular solution) at $\tau = T_s + 0.4\pi = 10.4\pi$. Panels (b) and (c) show an onset around $T_s + 0.8\pi = 20.8\pi$ and $T_s + 0.6\pi = 30.6\pi$, respectively. As a result of the time lag, the correlation coefficient for the solution curves in any two panels is low, although the solutions are regularly oscillatory. The results not only indicate a challenge in predicting the onset of oscillatory solutions but also suggest the importance of removing phase differences for verification (to obtain a better correlation between two solution curves).

For the three solution curves (which represent the same trajectory at different time intervals) in Figure 6, different times for the onset of solutions suggest a time lag (or phase differences). However, the period and amplitude for solutions within the regime of limit-cycle solutions are comparable, suggesting an optimistic view in predictability. Similar to Figure 6, Figure 7a,b provides the same trajectory during three different time intervals of $\tau \in [T_s, T_s + \pi]$, including $T_s$ for $12\pi$, $22\pi$, and $32\pi$. Such a choice is required in order to assure the full development of nonlinear oscillatory solutions. The selected time intervals are referred to as Epoch-1, Epoch-2, and Epoch-3, respectively, and the solution curve from the first epoch (i.e., Epoch-1) is used as a reference curve. The top panels clearly show phase differences amongst the solution curves, although the heating function remains the same for selected epochs. For solution curves during two different epochs, their cross correlation is computed to determine a time lag. Such a time lag is then considered to have a different starting time for the first epoch. For example, in panel (c), a time lag of $\Delta\tau = 0.0522$ is added to become a new starting time for the revised first epoch, referred to as a revised Epoch-1. After the time lag is considered, the solution curves for revised Epoch-1 and Epoch-2 are the same. In a similar manner, panel (d) displays the same evolution for solution curves for another revised Epoch-1 and Epoch-3.

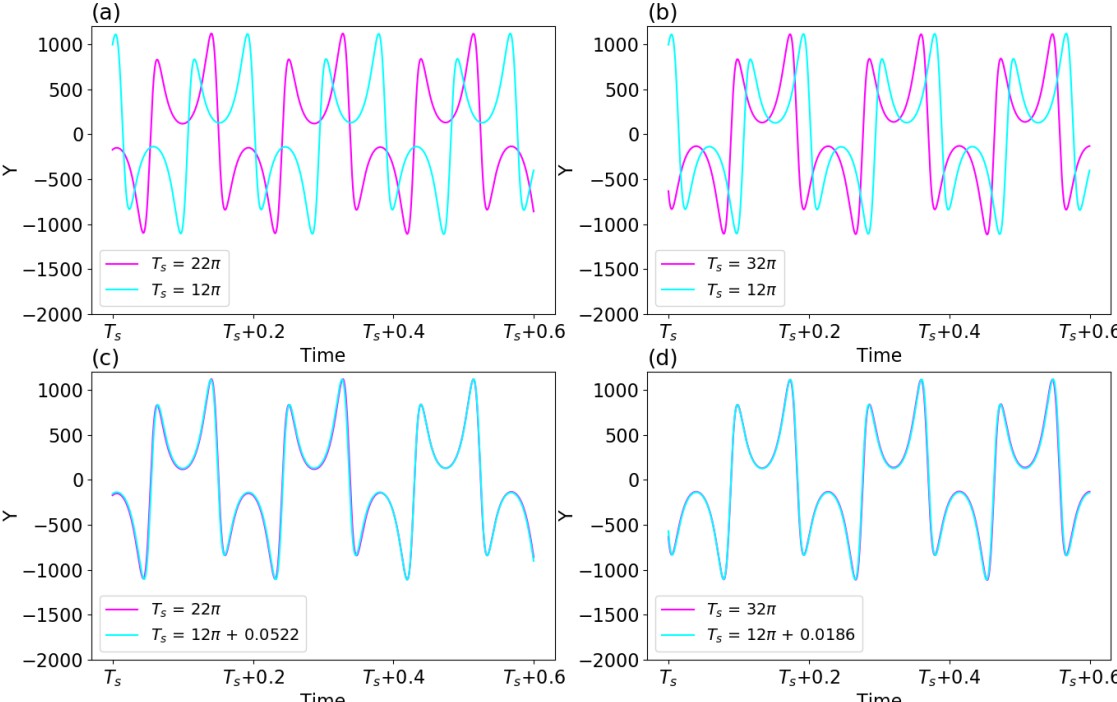

**Figure 7.** Panels (**a**,**b**) display the same trajectory during three different time intervals of $\tau \in [Ts, Ts + \pi]$, with the starting time $Ts$ equal to $12\pi$, $22\pi$, and $32\pi$, respectively. These three time intervals are referred to as Epoch-1, Epoch-2, and Epoch-3, respectively. In panels (**c**,**d**), to adjust the phase differences between two solutions curves, a time lag is added into Epoch-1.

Time varying multistability associated with the alternative occurrence of two kinds of attractor coexistence yields the alternative and concurrent appearance of various types of solutions. Such a feature indicates complexities of weather and climate that possess both chaotic and non-chaotic solutions, and their transitions. While chaotic solutions and their transition to regular solutions possess limited predictability, slowly varying solutions and nonlinear oscillatory solutions (i.e., limit-cycle type solutions) may have better predictability. The relationship between time varying multistability and distinct predictability is further elaborated after a discussion of the L69 model and its solutions.

### 2.6. Various Types of Solutions within the L69 Model

The L63 model is effective in revealing the chaotic nature of weather and climate and in suggesting a finite predictability [17]. The model has also been applied for estimating the predictability horizon using the global or finite-time Lyapunov exponent [48,49]. However, only a qualitative description of finite predictability has been established. The time varying, finite-time Lyapunov exponents suggest the dependence of finite predictability on initial conditions [9,50,51]. Some initial conditions may lead to better predictability than other initial conditions.

For a real-world application to predictability within the atmosphere, the L69 model was applied in order to produce a predictability limit of two weeks [47,52]. To examine the validity of the findings, a recent study by Shen et al., 2022 [17] summarized the major features of the L69 model, as follows:

- The L69 model is a closure-based, physically multiscale, mathematically linear, and numerically ill-conditioned system.
- As compared to other turbulence models, the L69 model applied the common assumptions of homogeneity and isotropy [53–61]. However, since the L69 model was derived from a conservative partial differential equation without dissipations, it is not a turbulence model.
- The L69 multiscale model has been used for revealing energy transfer and scale interaction.
- The L69 linear model cannot produce chaos.
- Since it possesses both positive and negative eigenvalues with large variances, yielding a large condition number (e.g., Figure 4 and Figure 5 of [17]), the L69 model produces a different kind of sensitivity, as compared to SDIC within the L63 model.
- The model permits the occurrence of linearly stable and unstable solutions as well as oscillatory solutions. However, only unstable solutions have been a focus in predictability studies.

Since the Lorenz 1969 multiscale model consists of 21, 2nd order ODEs (i.e., 42, 1st order ODEs), it is not necessarily viewed as a low-order dynamical system, as compared to many low-order systems in nonlinear dynamics textbooks. As compared to real world weather or climate models [19], it is a low-dimensional system. Given the above features of the L63 and L69 models, estimates for the predictability limit using either of the above models should be cautiously interpreted and should not be generalized as an upper limit for atmospheric predictability. Although these idealized models may not be suitable for quantitative analyses, as discussed below, the abundant features of their solutions may effectively, qualitatively reveal the complicated nature of weather and climate.

### 2.7. Distinct Predictability within Lorenz Models

As discussed, the L63 and L69 models have been applied for revealing the chaotic and unstable nature of weather and climate, and their intrinsic predictability, as well as the practical predictability of models [17,37]. Major findings regarding predictability and multistability within the two classical models and the GLM are summarized as follows:

- The L63 nonlinear model with monostability is effective for revealing the chaotic nature of weather, suggesting finite intrinsic predictability within the chaotic regime of the system (i.e., the atmosphere).

- The L69 linear model with ill-conditioning easily captures unstable modes and, thus, is effective for revealing the practical finite predictability of the model.
- The GLM with multistability suggests both limited and unlimited (i.e., up to a system's lifetime) intrinsic predictability for chaotic and non-chaotic solutions, respectively.

All of the above three systems in bullets are autonomous. Namely, system parameters are not explicit functions of time. By comparison, when a time varying heating function was applied within the GLM, the system became non-autonomous. Such a system produces a time varying multistability that is modulated by large-scale time varying forcing (heating). As a result, various types of solutions (e.g., steady-state, chaotic, and limit cycle solutions) with distinct predictability appear alternatively and/or concurrently. For example, slowly varying solutions may coexist with chaotic solutions for one period of time, and with limit cycle solutions for another period of time. Additionally, as illustrated in Section 2.5, the onset of an emerging solution (i.e., during a transition from a chaotic solution to a limit-cycle type solution) suggests a different predictability problem. In summary, the above discussions support the revised view that emphasizes the dual nature of chaos and order.

### 2.8. Non-Chaotic Weather Systems

The L69 model with forty-two, first order ODE was applied in order to study the multiscale predictability of weather. Although the L69 model is neither a low-order system nor a turbulence model (due to the lack of dissipative terms), major findings using the L69 model were indeed supported by studies using turbulence models [17,55,56]. By comparison, the L63 model has been used to illustrate the chaotic nature in weather and climate. Finite-dimensional chaotic responses revealed using the simple L63 model can be captured using rotating annulus experiments in the laboratory [62,63], illustrated by an analysis of weather maps (Figures 10.6 and 10.7 in [63]), and simulated using more sophisticated models (e.g., [64]). Based on ensemble runs using a weather model, the feature of local finite dimensionality [65–67] indicates a simple structure for instability (e.g., within a few dominant state space directions) (personal communication with Prof. Szunyogh) and, thus, suggests the occurrence of finite-dimensional chaotic responses.

As indicated by the title of Chapter 3 in [3], "Our Chaotic Weather", and the title of [63], "Application of Chaos to Meteorology and Climate", applying chaos theory for understanding weather and climate has been a focus for several decades [68,69]. By comparison, non-chaotic solutions have been previously applied for understanding the dynamics of different weather systems, including steady-state solutions for investigating atmospheric blocking (e.g., [70,71]), limit cycles for studying 40-day intra-seasonal oscillations [72], quasi-biennial oscillations [73] and vortex shedding [74], and nonlinear solitary-pattern solutions for understanding morning glory (i.e., a low-level roll cloud, [75]). While additional detailed discussions regarding non-chaotic weather systems are being documented in a separate study, Table 2 provides a summary.

**Table 2.** Non-chaotic Solutions vs. Weather Systems.

| Type | Weather Systems | References |
|---|---|---|
| Steady-state Solutions | Atmospheric blocking | [70,71] |
| Limit Cycles | 40-day intra-seasonal oscillations | [72] |
| | Quasi-biennial oscillations | [73] |
| | Vortex shedding | [74] |
| Nonlinear solitary-pattern solutions | Morning glory | [75] |

While the conventional view focuses on chaos, as well as monostability, our revised view emphasizes the possibility for coexisting chaotic and non-chaotic weather systems, yielding the concept of multistability. As illustrated using the analogy with skiing and kayaking in Figure 1 (within a physical space), the major difference between monostability and multistability in the acknowledgement of "inhomogeneity" (within a phase

space). Namely, SDIC (i.e., chaos) only appears within some physical/phase spaces instead of the entire spaces. Furthermore, the GLM with a time varying heating function, the so-called non-autonomous system [32,36], illustrated not only the temporal evolution but also the onset of various kinds of solutions, better representing the complexities of weather and climate, as compared to classical models with time independent parameters (i.e., autonomous systems).

### 2.9. Suggested Future Tasks

Numerous interesting studies in nonlinear dynamics exist and many of them have the potential to improve our understanding of weather and climate. Here, since suggested future tasks and additional studies and concepts will be covered in the future, only a few studies are discussed. Based on the above discussions, an effective classification of chaotic and non-chaotic solutions may identify systems with better predictability. Such a goal may be achieved by applying or extending existing tools, including the recurrence analysis method, the kernel principal component analysis method, the parallel ensemble empirical mode decomposition method, etc. (e.g., [45,46,76,77]). On the other hand, to separate chaotic and non-chaotic attractors, the detailed attractor basin for each attractor should be determined. Then, it becomes feasible to address final state sensitivity and intra-transitivity, as defined in Table 1. Whether or not the number of attractors in our weather is finite is another interesting but challenging question. Given a specific system that possesses infinite attractors, the detection of "megastability" and "extreme multistability" that correspond to countable and uncountable attractors [39], respectively, further increases the level of the challenge. In addition to the above three types of solutions and two kinds of attractor coexistence, homoclinic phenomena (e.g., homoclinic bifurcation, homoclinic tangencies, and homoclinic chaos), which have been intensively studied within low-order systems deserve to be explored in high-dimensional systems (e.g., using the GLM with nine modes or higher) [78–80].

There is no doubt that the "butterfly effect", originally derived from the Lorenz's 1963 study [1,3], is a fascinating idea that has inspired many researchers to devote their time and effort to related research. For example, Lorenz's attractor and Lorenz-type attractors (that are associated with Lorenz and Shilnikov types of saddle points, respectively) [1,3,81–85] have been rigorously examined within dynamical systems (e.g., the L63 model and the Shimizu-Morioka model [81]). Among the three kinds of butterfly effects classified by Shen et al., 2022 [47] and reviewed above, the first kind of butterfly effect with SDIC is well accepted, while the second kind of butterfly effect remains a metaphor. Thus, there is a definite need to finally answer the following question: "Can a butterfly flap cause a tornado in Texas?". Stated more robustly, can a small perturbation create a coherent larger scale feature at large distances? If so, what are the sizes and intensity that can produce such a feature and at what distances? All of the above questions are the subject of future studies.

## 3. Concluding Remarks

In this study, we not only provided a review of the features of various types of solutions within classical and generalized Lorenz models, but also presented a new analysis for the onset of emerging solutions, as well as a list of non-chaotic weather systems, to propose a refined version of the revised view:

> "The atmosphere possesses chaos and order; it includes, as examples, emerging organized systems (such as tornadoes) and time varying forcing from recurrent seasons",

in contrast to the conventional view of "weather is chaotic". The revised view focuses on: (1) the dual nature of chaos and order with distinct predictability in weather and climate; (2) the role of slowly varying environmental forcing (e.g., heating) in modulating the appearance of chaos and order, and in determining slowly varying systems; and (3) the challenge in determining the exact timing for the onset of a new, recurrent epoch (or regime).

From the perspective of dynamical systems, time varying multistability represents the key concept of the revised view. As compared to monostability that indicates the

appearance of single type solutions (such as steady-state, chaotic, or limit cycle solutions), multistability permits the coexistence of chaotic and non-chaotic solutions that occupy different portions of the phase space. To reveal fundamental differences between monostability and multistability, an analogy using skiing and kayaking was provided. In addition to diurnal and annual cycles, a short list of non-chaotic weather systems was provided in Section 2.8. Suggested future tasks were presented in Section 2.9.

**Author Contributions:** Conceptualization, B.-W.S., R.P.S., X.Z. (Xubin Zeng), and R.A.; methodology, B.-W.S.; software, B.-W.S., J.C., W.P., and S.F.-N.; writing—original draft preparation, B.-W.S., R.P.S., and X.Z. (Xubin Zeng); writing—review and editing, B.-W.S., R.P.S., X.Z., J.C., S.F.-N., W.P., A.K., X.Z. (Xiping Zeng) and R.A. All authors have read and agreed to the published version of the manuscript.

**Funding:** This research received no external funding.

**Institutional Review Board Statement:** Not applicable.

**Informed Consent Statement:** Not applicable.

**Data Availability Statement:** Not applicable.

**Acknowledgments:** We thank academic editors, editors, four anonymous reviewers, and Istvan Szunyogh for valuable comments and discussions.

**Conflicts of Interest:** The authors declare no conflict of interest.

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
