# Peer review of "The Dual Nature of Chaos and Order in the Atmosphere"

_atmosphere, doi:10.3390/atmos13111892_

Round 1

Reviewer 1 Report

This is the third revision of the manuscript.

I think that the manuscript is not suitable for publication because the authors claim to have done a description which is rather incomplete.

I recommend that the authors do not announce so strong results (description) because they are false. For readers that are unfamiliar with the topic, the description could be nice. But, for me (who have studied the Lorenz attractor for years), this article is not useful and contains false statements. 

For instance, do you know which role played Warwick Tucker to prove rigorously that the Lorenz model (geometrical one) contains a strange attractor twenty years ago? This important question has been ignored. 

As I said, although the images seem to be nice, this type of work does not contribute to the development of science. Since it claims to be a description of the (big) topic, it contains misinformation and false sentences. 

I recommend outright rejection.

Author Response

Please see the attached pdf file.

Reviewer 2 Report

The paper is improved. However, the authors insist on claiming that the features found in low-dimensional atmospheric models are features of the Earth's atmosphere.

Physicists usually claim the properties of a given natural system, like the atmosphere, based on data analysis of the system itself (data inferred from its observation). The manuscript discusses models, not observations. In my opinion, the authors should explicitly attribute the features they found in low-dimensional atmospheric models to these models, and not to the Earth's atmosphere, because they do not infer the features from their atmospheric observations. 

For example, the title should be something like this:

"The Dual Nature of Chaos and Order in the Atmospheric Low-dimensional Models"

and the same is within all the text of this manuscript. 

Author Response

Please see the attached pdf file.

Reviewer 3 Report

I am satisfied with the revision

Author Response

Thanks very much!

This manuscript is a resubmission of an earlier submission. The following is a list of the peer review reports and author responses from that submission.

Round 1

Reviewer 1 Report

The manuscript “The Dual Nature of Chaos and Order in the Atmosphere” by Shen et al. proposes a review of simplified weather models like the Lorenz 1963 and 1969 models and a generalized Lorenz model. 

The paper is interesting but is not presented satisfactorily. There are two main points on which the authors should work, namely:

  1. Throughout the paper, the authors confuse the properties of the low-dimensional models with those of the natural system to which the models are devoted. For instance, the authors found the coexistence of various attractors in the model they discussed, and this coexistence is a property of the model. As a result, the authors should clarify that the features they found are features of the models.
    (To infer features of the atmosphere, we would need to analyze data from atmospheric observations. However, this paper is about models and not atmospheric observations).
  2. The low-dimensional models such as Lorenz's 1963 and 1969 models designed for weather should also be discussed with respect to turbulence models and direct numerical simulations of the Earth's atmosphere.

Following are other criticisms and minor points:

  1. Title: The dual nature is not a dual nature of chaos and order. The model discussed in the article shows a dual nature as it admits both chaotic solutions and regular oscillatory solutions. The authors should clarify in the title that the dual nature is referring to models.
  2. Lines 40, 41. In the manuscript, “As reviewed in Section 2, both also produce…”
    For the sake of clarity, the authors should specify what “both” refers to. Both models?
  3. Lines 65, 66. The text in italic characters should always quote the source.
  4. Lines 72, 73. The analogy of monostability and multistability using skiing vs. kayaking should be described at the beginning of the paper, i.e., as soon as it is mentioned. Presenting this analogy at the beginning of the manuscript can help to understand the rest of the article better. For instance, this analogy may be more helpful at the beginning of section 2 and not at the end as it is now. In addition, the authors can recall this analogy when it is useful, e.g., when the text refers to multistability (e.g., from line 172 to line 177).
  5. In subsection 2.1, for the benefit of the readers, the authors should clarify the math of model L63, even if L63 is a very famous model. For instance, the authors mentioned three parameters of the model (one is the Rayleigh number, and the other two are considered constants). It is not clear where these parameters enter into the model. Indeed, the lack of the model equations makes the text not easily understood.
  6. Lines 90, 91. For the sake of clarity, can the authors write: “The only difference between the control simulation and the parallel simulation is adding the perturbation epsilon = 10^{-10} into the latter type of simulation”?
    Moreover, where is the perturbation added? Which physical quantity is perturbed?
  7. Lines 141, 142. Can the authors define “the aggregated negative feedback” and “the negative feedback”?
  8. Line 157. Can the authors further clarify the model (equations)? In particular, where does the heating parameter come into play?
  9. Line 213. The last sentence among brackets, namely “(After Shen et al., 2021 [3]).” is unclear. 
  10. Line 220. Can the authors provide more details on “the first appearance of African Easterly waves during a seasonal transition”?
  11. Lines 235, 236. In the manuscript, “… the same time evolution of oscillatory solutions should appear in the panels.”
    For clarity, can the authors specify the panels they are referring to?
  12. From line 239 to line 243. In the manuscript, “As a result of the time lag …”.
    For the benefit of the readers, can the authors explain more clearly why evaluating the correlation coefficient of the curves depicted in Fig. 5 is important?
  13. Line 249. In the text “in Figure 5, different times for the onset of solutions”
    The onset of solutions? Or onset of regular oscillations? 
  14. Line 272. In the text, “ Such a feature indicates complexities of weather and climate…” Can the authors change this sentence with the following:
    “Such a feature indicates complexities of the model we are discussing…”?
  15. Liens 279, 280. In the text, “The L63 model is effective in revealing the chaotic nature of weather and climate and suggesting a finite predictability”.
    Can the authors change this sentence with the following:
    “The L63 model reveals its chaotic nature and finite predictability”?
  16. Line 306. In the text “Given the above,…”. It is unclear what the authors are referring to.
  17. Liens 310, 311, 312. In the text, “the L63 and L69 models have been applied for revealing the chaotic and unstable nature of weather and climate, and their intrinsic predictability, as well as the practical predictability of models.”
    The authors should clarify that models show their features, i.e., features of the models. The features of the natural systems are revealed by their observation (observational data analysis). Please, see main point 1. 
  18. Lines 359, 360, 361, 362. In the manuscript is written, “including steady-state solutions for investigating atmospheric blocking (e.g., [57, 58]), limit cycles for studying 40-day intra-seasonal oscillations [59], quasi-biennial oscillations [60] and vortex shedding [61], and nonlinear solitary-pattern solutions for understanding morning glory (i.e., a low-level roll cloud, [62]).”
    For the general public, the authors should provide more details on atmospheric blocking, the 40-day intra-seasonal oscillations, the quasi-biennial oscillations, etc.
  19. Lines 363, 364. In the text, “Such a goal may be achieved by applying or extending existing tools.” Can the authors specify the tools they are referring to?
  20. Line 368. The text says, “as well as a comprehensive literature review.”
    The authors should quote papers about turbulent models and numerical simulations of the Earth’s atmosphere and atmospheric phenomena (see main point 2) unless they specifically write that this paper is only about low-dimensional weather models.  

Reviewer 2 Report

This paper is about "The Dual Nature of Chaos and Order in the Atmosphere" [from title].

The authors describe the Lorenz models (1963 is the first model as far as I know), useful to analyse the chaotic nature of weather and climate and for estimating the atmospheric predictability limit. 

Recently, generalized Lorenz models suggested a revised view that “The atmosphere possesses chaos and order; it includes emerging organized systems (such as tornadoes) and time varying forcing from recurrent seasons”, in contrast to the conventional view of “weather is chaotic”.

To be honest, I do not understand the point of the paper.  The authors proposed a description which is rather short (just includes the Lorenz models), are ignoring other models and the mathematical theory that describe Lorenz model (singular hyperbolic sets). The russian literature (Shilnikov, 1965- 1968; Shilnikov and Turaev, 1995) on this type os models are missing (as well). 

I think that the present review is not enough to guarantee publication in this journal. The paper, as it stands, is very far from a publishable review on the topic.

I recommend outright rejection. 

Reviewer 3 Report

The figures should be improved for i) their quality, and ii) labels, units and legends used in the plots.

Reviewer 4 Report

This study provided a review of significant findings within classical and generalized Lorenz models, as well as a comprehensive literature review, to elaborate on the revised view in contrast to the conventional idea of “weather is chaotic.” An analogy using skiing and kayaking was provided to reveal fundamental differences between monostability and multistability further.

The overall level of the paper is good: even if it is pretty simple, it is well written, and some important considerations are highlighted. This paper has the potential to be accepted, but some important points have to be clarified or fixed before we can proceed and positive action can be taken. We here summarize these points:

  1. It is better not to include references and abbreviations in the abstract or definition section. Also, the long form of all abbreviations should be provided in their first use, not the next ones.
  2. The keyword numbers should be between 3 and 5. Please select the most important ones in indexing your work.
  3. All the reference numbers in the text should be linked to their corresponding information at the end of the manuscript. Consequently, clicking on the numbers in the text makes their detail accessible promptly. The same procedure should be done for Figures and Tables.
  4. I suggest including a table and comparing the previous works that have been done on modeling the weather and briefly explaining their main ideas.
  5. The Introduction section provides valuable information for the readers. Nevertheless, some information presented is not accurate. Authors should revise better and more the current literature in the field. Several studies were omitted, such as ‘multistable chaotic systems with hidden attractors’ and ‘mega-stable nonlinear chaotic systems.
  6. I recommend including the mathematical equations of each system before its simulation results. Moreover, the parameters and initial conditions should be included in both the text and the captions.
  7. The provided simulative results are not completely convincing to me. I suggest replotting them sharply.
  8. What do you mean by CDIC in the caption of Figure 1? It seems it is explained later that this is not reasonable. You should explain it before using the abbreviation.
  9. Part C of Figure 2 should be centered so that the overall shape of this figure will be a reversed triangle.
  10. Please explain how the time lags of Figure 6 are derived from their cross-correlation.

The paper can be reconsidered for publication if the above questions are answered and problems are fixed.
